# Tea Sprout Picking Point Identification Based on Improved DeepLabV3+

Chunyu Yan [1,2], Zhonghui Chen [1], Zhilin Li [1], Ruixin Liu [1], Yuxin Li [1,2], Hui Xiao [1], Ping Lu [3] and Benliang Xie [1,2,*]

1   College of Big Data and Information Engineering, Guizhou University, Guiyang 550025, China
2   Power Semiconductor Device Reliability Engineering Center of the Ministry of Education, Guiyang 550025, China
3   State Key Laboratory Breeding Base of Green Pesticide and Agricultural Bioengineering, Key Laboratory of Green Pesticide and Agricultural Bioengineering, Ministry of Education, Guizhou University, Guiyang 550025, China
*   Correspondence: blxie@gzu.edu.cn

**Abstract:** Tea sprout segmentation and picking point localization via machine vision are the core technologies of automatic tea picking. This study proposes a method of tea segmentation and picking point location based on a lightweight convolutional neural network named MC-DM (Multi-Class DeepLabV3+ MobileNetV2 (Mobile Networks Vision 2)) to solve the problem of tea shoot picking point in a natural environment. In the MC-DM architecture, an optimized MobileNetV2 is used to reduce the number of parameters and calculations. Then, the densely connected atrous spatial pyramid pooling module is introduced into the MC-DM to obtain denser pixel sampling and a larger receptive field. Finally, an image dataset of high-quality tea sprout picking points is established to train and test the MC-DM network. Experimental results show that the MIoU of MC-DM reached 91.85%, which is improved by 8.35% compared with those of several state-of-the-art methods. The optimal improvements of model parameters and detection speed were 89.19% and 16.05 f/s, respectively. After the segmentation results of the MC-DM were applied to the picking point identification, the accuracy of picking point identification reached 82.52%, 90.07%, and 84.78% for single bud, one bud with one leaf, and one bud with two leaves, respectively. This research provides a theoretical reference for fast segmentation and visual localization of automatically picked tea sprouts.

**Keywords:** DeepLabv3+; deep learning; semantic segmentation; picking point identification

## 1. Introduction

Premium tea is a general term for high-quality tea. The picking method of tea sprouts is a crucial factor in determining their economic value [1]. In general, the single bud, one bud with one leaf, and one bud with two leaves are the primary tender materials of premium tea [2]. Mechanical picking can automatically pick tea leaves, but they cannot select a specific picking point. Machine picking will also destroy the integrity of the sprouts, which will reduce the economic value of the premium tea [3]. Premium tea sprouts are picked nearly by manual means. However, this picking method is inefficient and costly and cannot meet the immense demand for premium tea in the market.

In recent years, computer vision-based robots have been applied in the automatic picking of premium tea sprouts, which is expected to improve picking efficiency and reduce picking costs [4]. Many scholars have researched this topic and obtained many excellent results. Zhao et al. performed RGB analysis of tea bud images and obtained distinct tea bud characteristics using HSI color conversion, and they used HSV spatial transformation for tea bud segmentation [5]. Qian et al. applied the jump connection in U-Net and contrastive-center loss function to the SegNet model and achieved good

results in tea shoot segmentation [6]. Qi et al. applied the Otsu to the traditional watershed algorithm. They found that the algorithm showed a better ability to identify tea buds in complex backgrounds [7]. Hu et al. proposed a discriminative pyramid network for semantic segmentation of tea geometries in natural fields, and the MIoU and PA values of the method were 84.46% and 94.48%, respectively [8]. The tea garden environment has complexities, such as uncontrolled light conditions and high similarity between sprouts and old leaves [9]. The aforementioned methods can separate tea sprouts from the old leaves under certain conditions, but they cannot accurately identify and locate the sprout picking point. This incapability limits the automatic picking of tea sprouts.

The development of computer vision technology has improved the performance of convolutional neural networks (CNNs) in locating objects in images with complex backgrounds. CNNs are perceptrons composed of millions of neurons. After they are trained, they can localize objects in images with nearly no processing. DeepLabV3+ network is a CNN that can be used for pixel-level object detection. The method has been widely applied in the field of agriculture. Peng et al. applied DeepLabV3+ with Xception to segment lychees, and the MIoU of this method reached 0.765 [10]. Song et al. utilized DeepLabV3+ with ResNet-101 to classify calyces, branches, and wires in orchards, which obtained MIoU of 0.694 for uniform weights and 0.480 for median frequency weights on the homemade kiwifruit canopy image dataset [11]. Ayhan and Kwan applied DeepLabV3+ with Xception to segment the forest, grassland, and shrubland in the Slovenia dataset, which achieved an IoU of 0.9086 for the forest, 0.7648 for grassland, and 0.1437 for shrubland [12]. Zhang et al. adopted DeepLabV3+ with ResNet-18 to segment trunk, leaves, and apple in a commercial "Fuji" apple orchard [13]. The method achieved 94.8%, 97.5%, and 94.5% pixel classification accuracy for trunk, apple, and leaf (background), respectively. These studies have shown that DeepLabV3+ can segment tea sprouts from the background. It can also separate single buds, one leaf, and two leaves. The segmentation results of DeepLabV3+ can be used to identify the coordinates of tea sprout picking points.

In this study, we propose a lightweight tea sprout segmentation network, termed MC-DM, to segment tea sprouts. The recognition of tea sprout picking points was achieved on the basis of the segmentation results of the MC-DM. The architecture of the MC-DM was improved from the DeeplabV3+. First, the detection efficiency is essential in tea garden picking tasks. MC-DM used the modified MobilenetV2 as the backbone network to reduce the calculation while fully extracting image features. Second, the densely connected atrous spatial pyramid pool (DASPP) module was introduced into the MC-DM to improve the network feature extraction capability. Finally, RGB color separation and the Shi–Tomasi algorithm were used to detect partial corner points of single bud, one leaf, and two leaves. The corner point with the lowest value of the vertical coordinate was identified as the picking point corresponding to a single bud, one bud with one leaf, and one bud with two leaves. The experimental results show that the segmentation accuracy of MC-DM is nearly the same as that of DeepLabV3+. However, the number of parameters MC-DM has is greatly reduced. After the segmentation results of MC-DM are applied to picking point recognition, the recognition accuracy also meets the requirements of actual premium tea picking. We provide a new method for realizing real-time detection under the circumstance of photographing in outdoor tea fields.

## 2. Materials and Methods

### 2.1. Dataset

The collection dates of the tea sprout images used in the experiment were from mid-March 2021 to mid-April 2021, the period of premium tea picking. The collection location was the tea garden in Guiyang City, China. We used Redmi Note 7pro mobile phone (20 MP + 5 MP + 2 MP rear triple camera) to capture tea sprout images, and the size of the original image was $4000 \times 3000$ pixels. Weather conditions included cloudy, overcast, and clear skies. The picking times were from 09:00 to 17:00. The sampled tea sprout data have wide background variation, which was convenient for strengthening the robustness

and generalization ability of the segmentation network. A total of 1110 images of tea sprouts were collected, which satisfied the data requirements for the pixel-based semantic segmentation.

Data augmentation, as a method of data preprocessing, plays an important role in deep learning. In general, effective data augmentation methods can avoid over-fitting and improve the robustness of the model [14]. The general methods of data augmentation are changing brightness, flipping, and adding noise. The experiments were conducted using manual data augmentation given the enormous labor cost of fully supervised training. As shown in Figure 1, the number of datasets increased from 1110 to 5550 by flipping horizontally, changing brightness, and adding Gaussian noise. All images were scaled to $640 \times 480$ pixels by the linear interpolation algorithm to improve the efficiency of model training. The expanded samples were divided into the training set, validation set, and test set according to the ratio of 6:2:2.

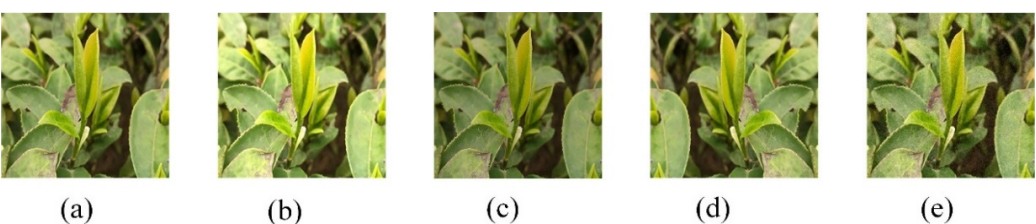

(a)     (b)     (c)     (d)     (e)

**Figure 1.** Selected samples of data enhancement. (**a**) Original image; (**b**) enhanced brightness; (**c**) reduced brightness; (**d**) horizontal flip; (**e**) noise addition.

### 2.2. Image Annotation

We manually labeled the tea sprout images with the LabelMe labeling tool into 4-pixel categories: background (cls0, black), single bud (cls1, red), one leaf (cls2, green), and two leaves (cls3, yellow). The original image is a 24-bit RGB image, and its corresponding visualization label is shown in Figure 2.

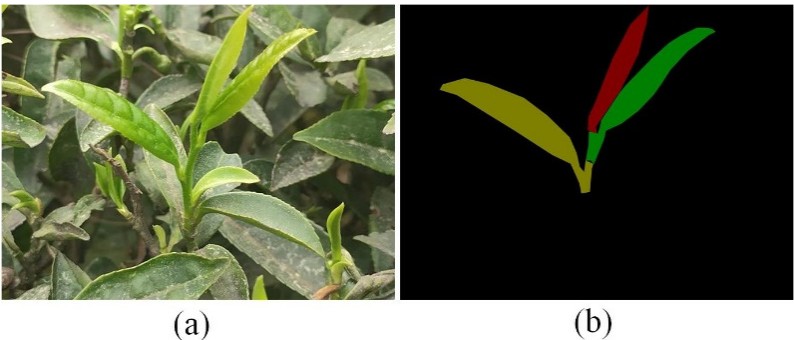

(a)     (b)

**Figure 2.** Tea sprout image and its corresponding visualization label. (**a**) Tea sprout image; (**b**) visualization label.

### 2.3. Description of Tea Sprout Picking Point

According to the different grades of premium tea and the frying process, the picking point of tea sprouts can be divided into single bud picking (A), one bud with one leaf picking (B), and one bud with two leaves picking (C). Among them, one bud with one leaf picking is the most extensive. Figure 3 shows the schematic of the corresponding standard picking point. The picking point is located at the center of the tea stalk connected with the tea sprout.

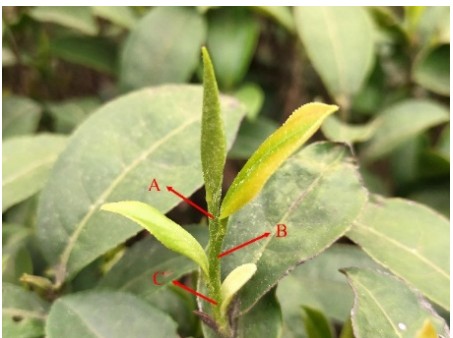

**Figure 3.** Picking point of tea sprout.

*2.4. Models*

2.4.1. MC-DM Architecture

Picking point recognition is a core technology of tea sprout picking systems [15]. The segmentation of single buds, one leaf, and two leaves are the prerequisite for identifying the picking point. DeepLabV3+ is a representative algorithm in the semantic segmentation field [16]. It includes two parts: an encoding module and a decoding module. In the encoding stage, the input image is first passed through the Xception backbone network to obtain a feature map of 16-time down-sampling. Then, the feature map of the 16-time down-sampling is placed into the atrous spatial pyramid pool (ASPP) module. The ASPP module consists of a $1 \times 1$ convolution, an average pooling layer with global information, and three $3 \times 3$ atrous convolutions with dilated rates of 6, 12, and 18. Finally, the feature maps obtained from the ASPP module are spliced, and the number of channels is compressed to 256 through a $1 \times 1$ convolution. In the decoding part, the feature map output from the encoding part is upsampled four times by a bilinear interpolation algorithm. Then, it is concatenated with the same-resolution feature maps extracted by the Xception backbone network. Finally, the tandem features are refined by $3 \times 3$ convolution, and the segmentation result is obtained by upsampling for four times. The DeepLabV3+ architecture is shown in Figure 4.

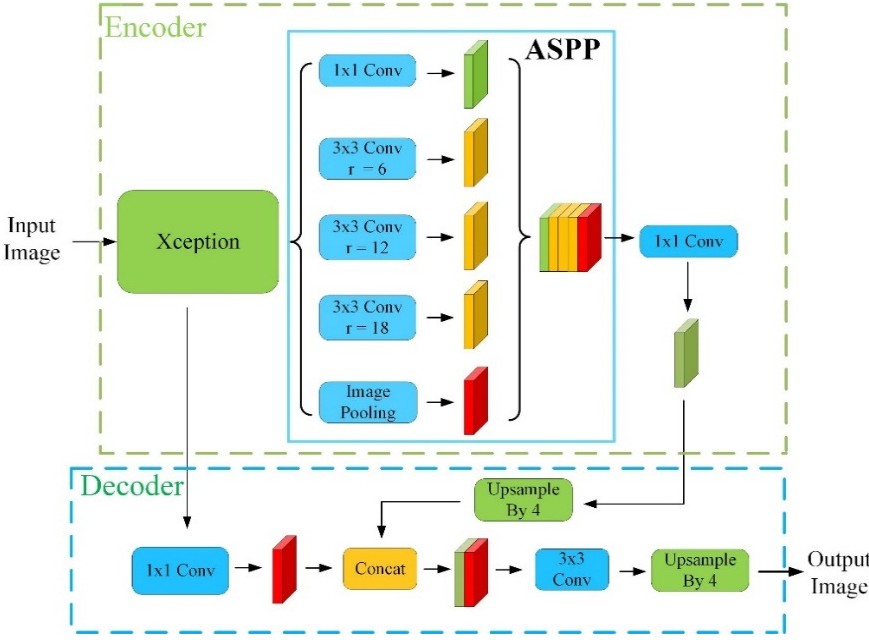

**Figure 4.** DeepLabv3+ network architecture.

MC-DM retains the excellent encoding–decoding structure of DeepLabV3+ to better identify tea sprouts. However, some improvements have been implemented. Figure 5 shows the network architecture of the MC-DM, and the improved parts are marked in red.

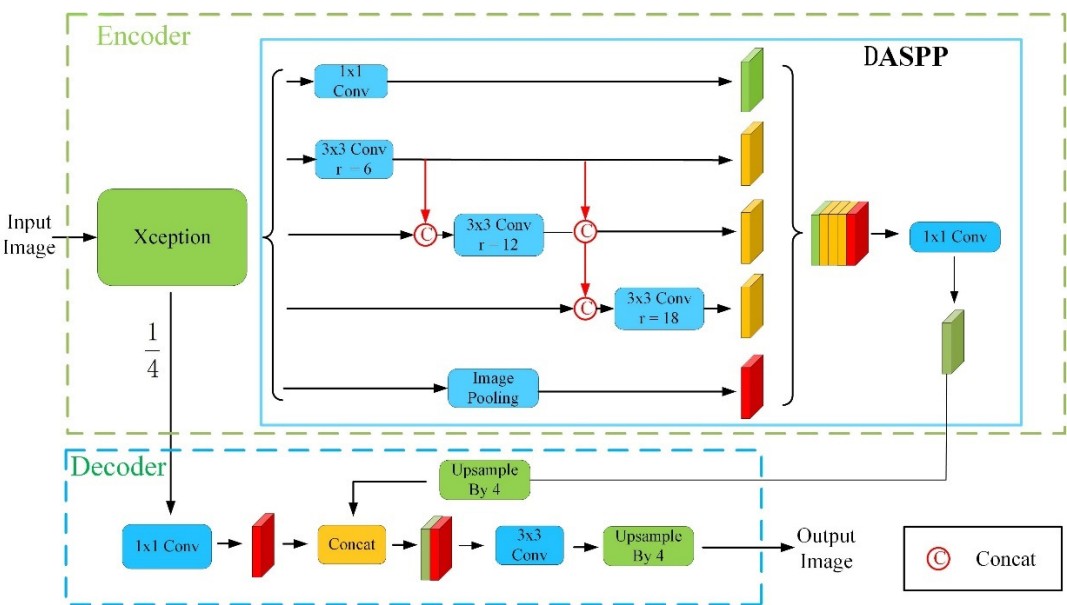

**Figure 5.** MC-DM network architecture.

As shown in Figure 5, MC-DM is an end-to-end network. The network input is a tea sprout image, and the output is a segmentation result of tea sprouts. In the encoding phase, the backbone network Xception is first replaced by a lightweight network. MobileNetV2 reduces the computational effort and mines the tea sprout features. The MobileNetV2 can extract feature maps of down-sampling for 2, 4, 8, and 16 times. Then, we place the feature map output of down-sampling for 16 times from the MobileNteV2 network into the DASPP. Compared with the atrous convolutions in the ASPP, those in the DASPP are cascaded by the series structure. The output of the atrous convolution with a smaller dilatation rate is contacted with the output of the backbone network. Then, they are sent to the atrous convolution with a larger dilatation rate to achieve a better feature extraction effect. In the decoding part, MC-DM retains the decoding network architecture of DeepLabV3+. The feature maps are restored to the original size of the input image through continuous upsampling.

### 2.4.2. Lightweight Backbone Network

MobileNetV2 is a lightweight network that can be embedded in mobile devices [17]. Its core contribution is to replace the standard convolution with deep separable convolution, which reduces the computational effort and model parameters. The deep separable convolution decomposes the standard convolution into two parts: deep convolution and point convolution. For the first time, the input features are first decomposed into multiple single-channel ones and convolved in each channel with a $3 \times 3$ convolution kernel to extract features, which is called deep convolution. Then, point convolution convolves the result of depth convolution with a $1 \times 1$ convolution to assemble the output features. Compared with standard convolution, the $3 \times 3$ deep separable convolution reduces the computation by 90% with only a slight reduction in inaccuracy [18]. On the basis of the deep separable convolution, the mobileNetV2 introduces an inverse residual structure, which further improves the network performance. In the inverse residual module, the input feature channels are first expanded by a $1 \times 1$ convolution. Then, the expended features are convolved with the $3 \times 3$ depth separable convolution to extract feature information. Finally, a $1 \times 1$ convolution is used to restore the number of feature channels. Compared

with the residual architecture, the inverse residual structure avoids the drawback that the network only has good extraction of low-dimensional features and loses information on high-dimensional features when the number of input feature channels is small [19]. The network architecture of the MobileNetV2 is shown in Table 1.

**Table 1.** MobileNetV2 network architecture.

| Input | Operator | $t$ | $c$ | $n$ | $s$ |
|---|---|---|---|---|---|
| $224 \times 224 \times 3$ | Conv2d | - | 32 | 1 | 2 |
| $112 \times 112 \times 32$ | bottleneck | 1 | 16 | 1 | 1 |
| $112 \times 112 \times 16$ | bottleneck | 6 | 24 | 2 | 2 |
| $56 \times 56 \times 24$ | bottleneck | 6 | 32 | 3 | 2 |
| $28 \times 28 \times 32$ | bottleneck | 6 | 64 | 4 | 2 |
| $14 \times 14 \times 64$ | bottleneck | 6 | 96 | 3 | 1 |
| $14 \times 14 \times 96$ | bottleneck | 6 | 160 | 3 | 2 |
| $7 \times 7 \times 160$ | bottleneck | 6 | 320 | 1 | 2 |
| $7 \times 7 \times 320$ | Conv2d $1 \times 1$ | - | 1280 | 1 | 1 |
| $7 \times 7 \times 1280$ | $7 \times 7$ Avgpooling | - | | 1 | - |
| $1 \times 1 \times 1280$ | $1 \times 1$ | - | k | - | - |

The input represents the size of the input feature map of the current layer. The operator represents the operations performed by MobileNetV2, including the normal convolution layer, the inverse residual structure, and the average pooling layer. $t$ is the expansion multiple of the channel in the inverse residual structure, $n$ denotes the number of repetitions for the current layer, $c$ is the number of output channels, and $s$ is the stride of the convolution in the current layer.

In this study, the modified MobileNetV2 is used as the MC-DM backbone network, and its architecture is shown in Table 2. First, the average pooling layer and the fully connected layer of MobileNetV2 are removed given that the semantic segmentation task needs to preserve the location information of image pixels. Second, only the first eight convolutional layers of MobileNetV2 are retained to extract features for reducing the computation and memory consumption of MC-DM. The original MobileNetV2 is aimed at the image classification task, and the size of the output feature map in the eighth layer is 1/32 of the original map. The stride size of convolution in this layer is changed to 1 to retain more feature information of tea sprouts. Furthermore, the size of the output feature map in this layer is increased to 1/16 of the original map.

**Table 2.** Backbone network architecture.

| Input | Operator | $t$ | $c$ | $n$ | $s$ |
|---|---|---|---|---|---|
| $640 \times 480 \times 3$ | Conv2d | - | 32 | 1 | 2 |
| $320 \times 240 \times 32$ | bottleneck | 1 | 16 | 1 | 1 |
| $320 \times 240 \times 16$ | bottleneck | 6 | 24 | 2 | 2 |
| $160 \times 120 \times 24$ | bottleneck | 6 | 32 | 3 | 2 |
| $80 \times 690 \times 32$ | bottleneck | 6 | 64 | 4 | 2 |
| $40 \times 30 \times 64$ | bottleneck | 6 | 96 | 3 | 1 |
| $40 \times 30 \times 96$ | bottleneck | 6 | 160 | 3 | 2 |
| $40 \times 30 \times 160$ | bottleneck | 6 | 320 | 1 | 1 |

We do not randomly initialize the weights of the MC-DM model but instantiate the pre-trained weight of ImageNet in the modified MobilenetV2 network. ImageNet's pre-trained weight has shown extraordinary achievements in the image analysis field. It contains over 14 million images of different scenes. During optimization, migrating the pre-training weights of ImageNet can fit the pre-training model to a specific region of interest, which accelerates the model convergence and optimizes the model weights.

### 2.4.3. DASPP

DeepLabV3+ uses the ASPP module that contains atrous convolution to extract the multi-scale features of the input image for retaining the detailed features of image convolution and increasing the receptive field. When the size of the convolution kernel is *k* with a dilation rate of *r*, the receptive field of the atrous convolution in the ASPP module is

$$R = (r - 1) \times (k - 1) + k \tag{1}$$

As shown in Equation (1), the receptive field of the atrous convolution is proportional to its dilation rate. The atrous convolution expands the convolution receptive field by filling zeros in the convolution kernel and outputs the convolution results of nonzero sampling points. As the dilation rate increases, the nonzero pixel sampling of the atrous convolution becomes sparser. The information obtained by atrous convolution is seriously lost under the same computing conditions. This situation is detrimental to the learning and training of the model.

The ASPP module of the DeepLabV3+ network is improved through dense connection by referring to the DenseNet network to solve the problem of information loss due to the enlarged receptive field of the ASPP module. The network architecture of DASPP is shown in the yellow dashed box in Figure 5. Compared with those of the ASPP in Figure 4, the three atrous convolutions of DASPP are cascaded by the series structure. The output of the atrous convolution with less dilation rate is contacted with the resulting output from the backbone network. Then, it is sent to the atrous convolution with more dilation rate to achieve a better feature extraction effect. The process is illustrated by an atrous convolution with a dilation rate of 12. Figure 6a shows the pixel sampling of the atrous convolution with a dilation rate of 12 in the ASPP. The receptive field of the atrous convolution is 25, and the number of elements involved in the computation is only 9. Figure 6b shows the pixel sampling of the atrous convolution with a dilation rate of 12 in DASPP. The number of pixels involved in the computation is increased from 3 to 7. The receptive field is raised from 25 to 49.

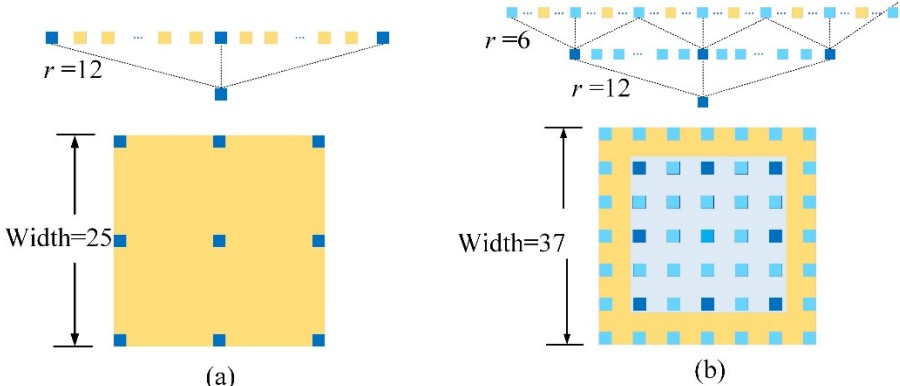

**Figure 6.** Distribution of atrous convolution sampling before and after dense connection. (**a**) Before dense connection; (**b**) After dense connection.

For the computation of the receptive field, DASPP can provide a larger receptive field. $R_{k,r}$ denotes the receptive field provided by the atrous convolution with the kernel size *k* and dilation rate *r*. The atrous convolutions in the ASPP module work in parallel and do not share any information. Therefore, the maximum receptive field of ASPP is the maximum of the receptive fields provided by each atrous convolution. From Equation (2), the maximum receptive field of the ASPP module is

$$R = \max(R_{3,6}, R_{3,12}, R_{3,18}) = 37 \tag{2}$$

A larger receptive field can be obtained by cascading two atrous convolutions. The calculation of the cascaded receptive field is shown in Equation (3), where $R_1$ and $R_2$ are two different receptive fields of atrous convolution, respectively.

$$R = R_1 + R_2 - 1 \tag{3}$$

According to Equation (3), the maximum receptive field of DASPP is

$$R = \max(R_{3,6} + R_{3,12} + R_{3,18}) = 73 \tag{4}$$

We define information utilization $\beta$ to measure the relationship between receptive field and information utilization $\beta$. The calculation of $\beta$ is shown in Equation (5), where $X1$ is the number of elements in the receptive field involved in computation and $X2$ is the total number of elements in the receptive field. Table 3 shows the performance of the atrous convolution with different connections in the feature map.

$$\beta = \frac{X1}{X2} \tag{5}$$

**Table 3.** Effect of ASPP connection method on the atrous convolution.

| ASPP Connection | Dilation | Receptive Field | Effective Elements | $\beta$ (%) |
|---|---|---|---|---|
| ASPP | 6, 12 | 25 | 9 | 1.44 |
| | 6, 12, 18 | 37 | 9 | 0.66 |
| | 6, 12, 18, 24 | 49 | 9 | 0.37 |
| DASPP | 6, 12 | 37 | 49 | 2.18 |
| | 6, 12, 18 | 73 | 255 | 4.79 |
| | 6, 12, 18, 24 | 122 | 961 | 6.46 |

As shown in Table 3, the atrous convolution in the ASPP module works in parallel, which ignores the correlation between the atrous convolution with different dilution rates. As the number of atrous convolutions increases, the pixel utilization decreases. By contrast, DASPP achieves information sharing between different atrous convolution branches, which increases the range of receptive fields and significantly improves pixel utilization.

The DASPP module increases the receptive field and reduces the information loss, but it increases the model parameters and decreases the model computation speed. A $1 \times 1$ convolution is added before the atrous convolution of DASPP to reduce the channel dimension of the input features, which decreases the model parameters, for solving the abovementioned problem. In this study, the modified MobilenetV2 is used as the backbone network. The input feature channel of the ASPP module is 320, and the output feature channel dimension is 256. Then, the parameter number of the ASPP module is calculated, as shown in Equation (6).

$$N_1 = 320 \times 256 \times 9 \times 3 = 2,211,840 \tag{6}$$

The number of input feature channels for the atrous convolution in the DASPP module is reduced from 320 to 256, and the output feature channel dimension is 256. The parameter number of the DASPP module is calculated as shown in Equation (7).

$$N_2 = (320 + 576 + 832) \times 256 + 256 \times 256 \times 9 \times 3 = 2,211,840 \tag{7}$$

As shown in Equations (6) and (7), the proposed DASPP maintains the same number of parameters as the ASPP.

#### 2.4.4. Picking Point Positioning Method

The quality of premium teas requires ensuring the integrity of the sprouts and limiting the picking area to the sprout stems. The identification process of tea sprout picking points is shown in Figure 7. First, three binary images of the same size as the original image need to be obtained. The reason is that the color difference among the background, single bud, one leaf, and two leaves of the divided tea buds is obvious and the internal color components are the same. Therefore, the binary operation for a single bud, one leaf, and two leaves can be completed according to the color range of the pixel points in the tea bud image. Then, the pixel points within the discrete area of the binary image are counted. This way determines the maximum connected area for single bud, one leaf, and two leaves, which removes the interference of background information on picking point recognition. Finally, the corner points of the maximum connected area are calculated using the Shi–Tomasi algorithm. The corner point with the lowest vertical coordinate is identified as the corresponding pick point. Table 4 shows the RGB color segmentation thresholds for a single bud, one leaf, and two leaves.

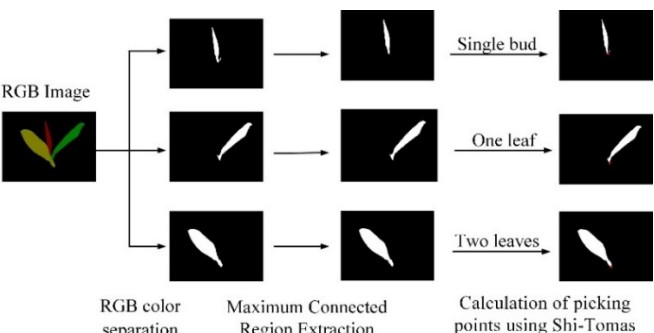

**Figure 7.** Picking point identification process.

**Table 4.** Segmentation thresholds of tea buds.

|  | Bud | One Leaf | Two Leaves |
|---|---|---|---|
| R | 255 | 0 | 255 |
| G | 0 | 255 | 255 |
| B | 0 | 0 | 0 |

#### 2.5. Evaluation Metrics

The experiment uses multiple levels of control parameter variables for the evaluation to more effectively measure the performance of each model. The main evaluation indexes include the accuracy of the model prediction, the prediction speed of the model, and the size of the model parameters. Many criteria are used to measure the accuracy of image segmentation. In general, MIoU is the most representative evaluation metric in the fields of image segmentation. IoU refers to the intersection between the predicted set of values and the actual set of values for a single class of pixels. MIoU is the average of IoUs of all categories. It reflects the ability of the model to segment the image pixels. The mathematical expression of MIoU is shown in Equation (8).

$$MIoU = \frac{1}{k+1} \sum_{i=0}^{k} \frac{p_{ii}}{\sum_{j=0}^{k} p_{ij} + \sum_{j=0}^{k} p_{ji} - p_{ii}} \tag{8}$$

where $k + 1$ is the number of total classes, including a background class; $p_{ii}$ is the number of pixels predicted to be correct. $p_{ij}$ is the number of pixels belonging to class $i$ but predicted to be class $j$. $p_{ji}$ is the number of pixels belonging to class $j$ but is predicted to be class $j$.

## 3. Results and Discussion

### 3.1. Implementation Details

We conducted experiments and validated our proposed method based on the PyTorch platform on the same workstation, which has the following configuration and installed software: Intel Core i7-11700@ 3.4 GHZ×8 threads, NVidia GeForce GTX 3080 GPU with RAM of 12 GB, 64-bit Windows 10 operating system, Python 3.8, OpenCV 3.4.1 and PyTorch1.4.1. During the model training process, tea sprout images were used as the input, and the stochastic gradient descent was used as the optimization function. All networks were trained for 800 iterations with a batch size of 16. The initial learning rate was set to 0.04 and multiplied by 0.5 every five training iterations. The momentum parameter is 0.9.

### 3.2. Backbone Network Validation

Considering the effectiveness of the Deeplabv3+ network used for semantic image segmentation, we apply it to the semantic segmentation of tea sprout images in this study. Tea picking robots have high requirements on the prediction speed of embedded models and the number of model parameters. Thus, Deeplabv3+ with a lightweight Mobilenetv2 backbone is designed as the base network. This section compares Deeplabv3+-Mobilenetv2 with the original Deeplabv3+, which is trained and tested under the same experimental conditions, to verify the effectiveness of the design choice. Their performance metrics are shown in Table 5.

**Table 5.** Comparison of the influence of different backbone networks.

| Method | Backbone | MIoU/% | Parameters/MB | Speed (f/s) |
|:---:|:---:|:---:|:---:|:---:|
| Original Deeplabv3+ | Xception | 91.33 | 104.61 | 25.10 |
| Deeplabv3+-Mobilenetv2 | MobilenetV2 | 90.33 | 11.18 | 41.14 |

Table 5 shows that DeepLabV3+ with different backbone networks has different detection results for the tea sprout dataset. The MIoU of the Deeplabv3+-Mobilenetv2 is 90.33%, the model prediction speed is 41.14 f/s, and the number of model parameters is 11.18 MB. The MIoU of the DeepLabV3+-MobilenetV2 model is only slightly reduced compared with that of the original DeepLabV3+ model. However, the DeepLabV3+-MobilenetV2 model has a 16.04 f/s improvement in model segmentation speed and an 86.13% reduction in the number of model parameters. By using MobilenetV2 instead of Xception as the backbone of the DeepLabV3+ model, the real-time performance and lightweight property of the algorithm have been significantly improved. This optimization is extremely beneficial for future deployment on mobile tea-picking devices. The design choice of MobileNetV2 as the backbone network is verified.

### 3.3. Analysis of DASPP and Dilation Rate Combinations

This section of the experiment is to verify the effect of the ASPP connection method and different dilation rates on the experimental results. Table 6 shows the effect of varying ASPP connection methods and dilation rate groups on MC-DM network performance when the backbone network is MobilenetV2. For the same expansion rate group, the segmentation speed and the number of parameters are essentially the same for the DASPP and ASPP models. However, the MIoUs of the DASPP model are higher than those of the ASPP model. The maximum improvement of the MIoU reaches 2.13%. These results show that the DASPP model with a larger receptive field can segment the tea sprout images more accurately, which validates the effectiveness of the DASPP connection.

**Table 6.** Comparison of the influence of different ASPP connection methods and dilation rates.

| ASPP Connection | Dilation Rates | Receptive Field | MIoU | Parameters/MB | Speed (f/s) |
|---|---|---|---|---|---|
| | 6, 12 | 25 | 85.47 | 6.75 | 45.05 |
| ASPP | 6, 12, 18 | 37 | 90.33 | 7.56 | 24.77 |
| | 6, 12, 18, 24 | 49 | 90.50 | 8.36 | 37.06 |
| | 6, 12 | 37 | 87.60 | 6.76 | 44.78 |
| DASPP | 6, 12, 18 | 73 | 91.85 | 7.63 | 40.82 |
| | 6, 12, 18, 24 | 122 | 92.02 | 8.56 | 36.89 |

In terms of dilation rate selection, the segmentation accuracy of the dilation rate DASPP(6,12) model is the lowest, and it has an MIoU of only 87.60%. The MIoU for the dilation rate DASPP(6,12,18) model and the dilation rate DASPP(6,12,18,24) model are 91.85% and 92.02%, respectively. The difference in MIoU values between them is 0.17. Thus, they can be considered to have the same segmentation accuracy. However, compared with the dilation rate DASPP(6,12,18) model, the model parameters of the dilation rate DASPP(6,12,18,24) model is increased by 0.93 MB, and the segmentation speed is decreased by 3.93 f/s. Therefore, we chose DASPP(6,12,18) as the combination of the atrous convolution dilation rate for MC-DM.

### 3.4. Comparison with Other Segmentation Models

We conducted comparative experiments between MC-DM and other segmentation models to further verify the effectiveness and feasibility of the MC-DM. Table 7 shows the evaluation results of different models for tea sprout segmentation.

**Table 7.** Evaluation results of different methods.

| Model | MIoU/% | Parameters/MB | Speed (f/s) |
|---|---|---|---|
| PSPNet | 83.23 | 29.93 | 40.85 |
| SegNet | 85.79 | 29.45 | 24.77 |
| U-Net | 86.29 | 13.40 | 25.79 |
| DeepLabV3+ | 91.33 | 54.52 | 25.10 |
| MC-DM | 91.85 | 7.63 | 40.82 |

As shown in Table 7, the MIoU of MC-DM reaches 91.85%, which is 8.62%, 6.06%, 5.56%, and 0.52% higher than those of PSPNet [20], SegNet [21], U-Net [22], and the DeepLabV3+, respectively. The MC-DM model has the highest segmentation accuracy. In terms of model lightweight property, the model parameter of MC-DM is only 7.63 MB, which is 74.51%, 74.09%, 43.06%, and 86.01% lower than those of PSPNet, SegNet, U-Net, and the DeepLabV3+, respectively. The MC-DM model performs optimally in terms of model lightweight property. In addition, the segmentation speed of the MC-DM model reaches 40.82 f/s, which is 16.05 f/s, 15.03 f/s, and 15.72 f/s better than those of SegNet, U-Net, and DeepLabV3+, respectively. The segmentation speed of the PSPNet model is the same as that of the MC-DM model. After the above-mentioned comparison, the MC-DM model achieves excellent performance in three metrics: segmentation accuracy, model lightweight, and segmentation speed.

Figure 8 shows the prediction results of different segmentation models for tea sprout images. The first and fifth columns are the original image, annotated image, PSPNet segmentation results, SegNet segmentation results, U-Net segmentation results, DeepLabV3+ segmentation results, and MC-DM segmentation results, respectively. The segmentation results of PSPNet are the worst. For images where the sprout color is similar to the background color, PSPNet nearly loses its segmentation ability. SegNet and U-Net segmentation results are better than PSPNet segmentation results. However, many pixel segmentation

errors are found in their segmentation results. MC-DM and DeepLabV3+ segmentation results are close to the annotated images. However, the segmentation results of the MC-DM model for tea sprouts are more refined.

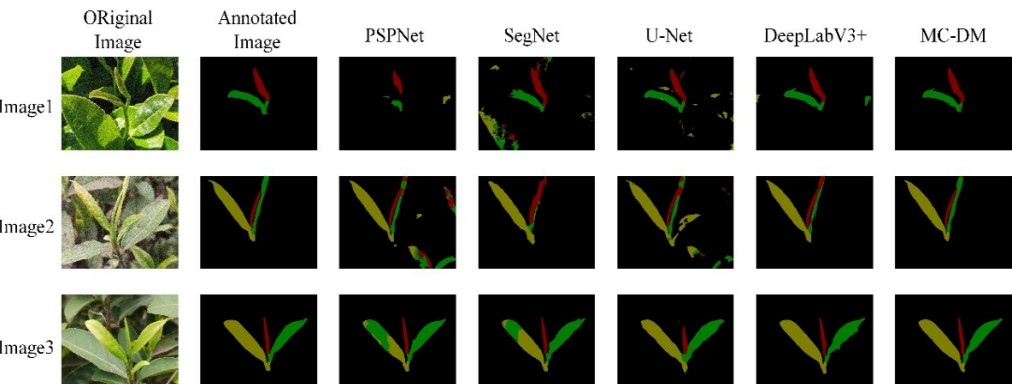

**Figure 8.** Prediction results of different segmentation models.

The comparison results in Table 7 and Figure 8 also show that the MC-DM model can achieve better tea sprout segmentation with lower computational resources. It can effectively segment the background, single bud, one leaf, and two leaves in the tea sprout image.

*3.5. Picking Point Marking Validation*

The two-dimensional coordinates of the tea sprout picking point are calculated using the tea images segmented with the trained MC-DM model and the coordinates are marked on the tea sprout pictures to further evaluate the segmentation effect of MC-DM and verify the effectiveness of the pick point localization method. When the picking point is located on the corresponding tea stalk, it is recorded as successful marking; otherwise, it is recorded as failed marking. Figure 9 shows the marking results of some picking points. Notably, the blue points are picking points for single bud, the green points are picking points for one bud with one leaf, and the red points are picking points for one bud with two leaves. Failed picking points are marked with a red frame. Our method has excellent results in locating the tea sprout picking points in the natural background. Most of the tea sprout picking points under different illumination are correctly marked.

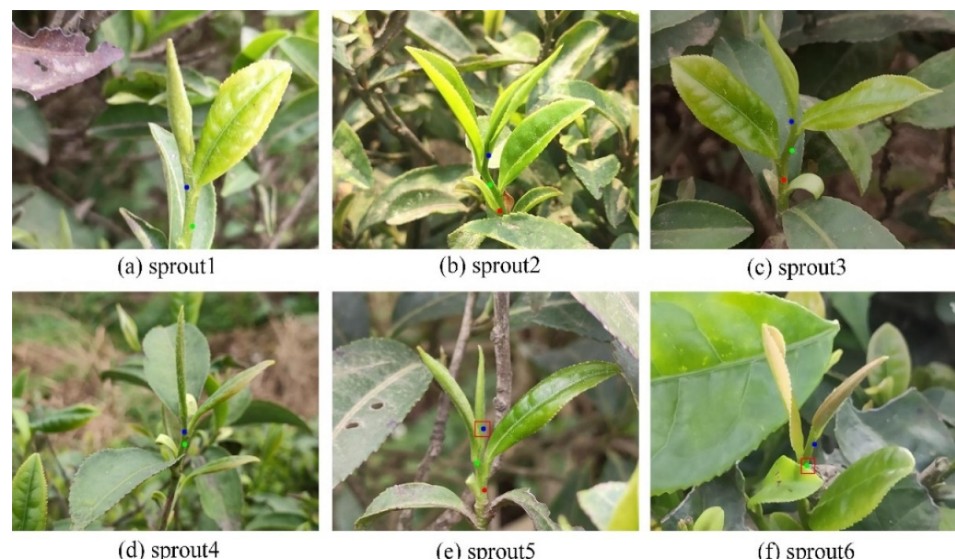

**Figure 9.** Results of picking point marking. (**a**–**d**) Correctly marked picking points; (**e**,**f**) Incorrectly marked picking points.

Tables 8–10 show the picking point identification statistics of 1100 tea sprouts. The identification accuracy of all three types of picking points is over 80%. The recognition accuracy of one bud with one leaf picking point reaches 90.07%. Considering that premium tea is mainly picked from one bud with one leaf, the proposed picking point identification method meets the requirements of the picking robot for the picking point identification accuracy.

**Table 8.** Identification results of picking point for single bud.

| Sample | 1 | 2 | 3 | 4 | 5 | Total |
|---|---|---|---|---|---|---|
| Number of picking points | 222 | 222 | 222 | 222 | 222 | 1110 |
| Number of successful identifications | 187 | 177 | 182 | 186 | 184 | 916 |
| Number of failed identifications | 35 | 45 | 40 | 36 | 38 | 194 |
| recognition accuracy/% | 84.23 | 79.73 | 81.98 | 83.78 | 82.88 | 82.52 |

**Table 9.** Identification results of picking point for one bud with one leaf.

| Sample | 1 | 2 | 3 | 4 | 5 | Total |
|---|---|---|---|---|---|---|
| Number of picking points | 221 | 221 | 222 | 222 | 222 | 1108 |
| Number of successful identifications | 201 | 199 | 193 | 201 | 204 | 998 |
| Number of failed identifications | 20 | 22 | 29 | 21 | 18 | 110 |
| recognition accuracy/% | 90.95 | 90.00 | 86.94 | 90.54 | 91.89 | 90.07 |

**Table 10.** Identification results of picking point for one bud with two leaves.

| Sample | 1 | 2 | 3 | 4 | 5 | Total |
|---|---|---|---|---|---|---|
| Number of picking points | 162 | 167 | 161 | 172 | 166 | 828 |
| Number of successful identifications | 132 | 149 | 136 | 145 | 140 | 702 |
| Number of failed identifications | 30 | 18 | 25 | 27 | 26 | 126 |
| recognition accuracy/% | 81.48 | 89.22 | 84.47 | 84.30 | 84.34 | 84.78 |

The main reasons for the failure of picking point identification are as follows. First, the MC-DM model has segmentation errors for sprout stems in some tea images due to the small target size of spout stems compared with the overall tea sprout. This condition leads to the failure in marking the corresponding picking points. Second, some of the single bud stems are wrapped by young leaves (Figure 9e), which leads to the incorrect identification of the picking point. It is the main reason for poor single bud picking point identification. The stems of one bud with two leaves are also located at the bottom of the tea sprout. It is easily obscured by old leaves (Figure 9f), which results in failure in picking point identification.

## 4. Conclusions

This study proposed a lightweight segmentation network based on DeeplabV3+, named MC-DM, to segment tea sprouts and identify the picking points. First, the lightweight MobileNetV2 replaced the original backbone network Xception to reduce the number of model parameters and improve the speed of model segmentation. Then, a densely connected spatial pyramidal pooling module was introduced into the MC-DM network, which enabled the network to obtain denser pixel sampling and a large receptive field. Finally, the tea sprout picking points were identified using the Shi–Tomasi algorithm based on the

segmentation results of the MC-DM network. The experimental results show that MC-DM achieves 91.58% of MIoU for tea sprouts. MC-DM has higher segmentation accuracy and fewer network parameters than other segmentation methods. In addition, the recognition accuracy rates of picking points for single bud, one bud with one leaf, and one bud with two leaves are 82.52%, 90.07%, and 84.78%, respectively. Its performance meets the requirements of the tea picking robot for picking point recognition accuracy.

**Author Contributions:** Conceptualization, C.Y. and Z.C.; methodology, C.Y.; software, C.Y. and Z.C.; validation, Z.C. and Z.L.; formal analysis, C.Y. and Z.L.; investigation, C.Y., Z.C. and Y.L.; data curation, R.L. and Y.L.; writing—original draft preparation, C.Y.; writing—review and editing, C.Y., Z.C.; visualization, R.L. and H.X.; supervision, P.L. and B.X.; project administration, P.L. and B.X.; funding acquisition, P.L. and B.X. All authors have read and agreed to the published version of the manuscript.

**Funding:** This research was funded by the National Natural Science Foundation of China (grant number 61562009), National Key Research and Development Program (grant number 2016YFD0201305-07), Open Fund Project in Semiconductor Power Device Reliability Engineering Center of Ministry of Education (grant number ERCMEKFJJ2019-06), and the Guizhou University Introduced Talent Research Project (grant number 2015-29).

**Institutional Review Board Statement:** Not applicable.

**Informed Consent Statement:** Not applicable.

**Data Availability Statement:** Not applicable.

**Conflicts of Interest:** The authors declare no conflict of interest.

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
