# Peer review of "Tea Sprout Picking Point Identification Based on Improved DeepLabV3+"

_agriculture, doi:10.3390/agriculture12101594_

Round 1
Reviewer 1 Report
This paper proposes a custom CNN to automatically identify tea sprouts and localize the picking point. The authors create MC-DM architecture based on the popular segmentation network DeepLabV3+. The custom CNN and the algorithm for localizing picking points are evaluated with a custom dataset.
Overall, I found the application of CNNs for tea sprout identification to be innovative and quite interesting. The authors thoughtfully break down the problem into individual small tasks suitable for machine vision. While the overall approach is innovative, the scientific soundness of the work needs to be significantly strengthened. In addition, the paper is poorly written in English, making it more difficult to judge the soundness of the work.
Below are my questions and suggestions:
1. What is the target application setup? Computer vision methods are known to suffer from complex uncontrolled environments. It is unclear what deployment scenario this paper is aiming at. Is the algorithm supposed to be run in a mobile robot outdoor in the tea fields? Or would this be used in a factory setting in which the environment is well-controlled? These different deployment scenarios have drastically different application requirements. The target application requirements need to be explicitly specified.
2. The instance segmentation problem is formulated in an unconvincing way. The authors labeled the tea sprout images into four mutually exclusive categories -- background, "single bud", "one leaf", and "two leaves". However, based on the example images shown in the paper, the "single bud", "one leaf", and "two leaves" categories are visually quite similar. In fact, even the background contains many leaves, which are visually similar to foreground pixels. The authors need to provide more evidence and explanation about why the problem can be formulated in this way.
3. The CNN network modification proposed in the paper is not well-justified and seems unnecessary for solving the problem. For example, the replacement of Xception with MobileNetV2 made the algorithm run ~15ms faster per frame in the authors' experiment. Why is that ~15ms speed improvement significant or needed? Furthermore, there is no description of the system setup for such speed measurement. Is the speed measured on a single-core CPU? Has the workload been parallelized? There are many factors that can impact the execution speed of a CNN network. The authors did not provide enough justification to demonstrate their changes to the CNN architecture made a significant difference.
4. The experimental setups and evaluation need to be described in a lot more detail. It's unclear how the authors trained the baseline model and their model. Which part of the model is fine-tuned vs using pre-trained weights? How are the hyperparameters tuned? What are the visual features the model actually picks up? What are the reasons and evidence behind the accuracy improvement in the modified CNN architecture? More detailed experiment descriptions and analyses are needed to make the work technically sound.
Author Response
请参阅附件。

Reviewer 2 Report
1) The authors should carry out a careful grammatical revision of the article, some texts could be improved to make them more understandable for the reader, specifically I suggest a technical linguistic revision, I present some particular cases, but in general a more detailed revision with a specialist is required.
- the backbone network to fully extract image features while reducing the calculation. After the above comparison 79-80
- much fewer 88
- Applying the segmentation results of MC-DM to picking point recognition, the accuracy of picking point recognition also meets the requirements of actual premium tea picking. 88-90 (rephase to avoid repeated picking point recognition)
- MC-DM model achieves excellent performance in three metrics: segmentation accuracy, model lightweight, and segmentation speed compared with the compared segmentation models. 370-372
2) In line 47 there is a typographical error that is relevant, "Ostu" should be "Otsu".
3) The authors should carefully check the numbers of their images, because in the case of the proposed method it is said to be figure 4, when the modification appears in figure 5.
However, some improvements have been implemented. Figure. 4 shows the network architecture of the MC-DM, and the improved parts are marked in red. 153-155
4) In figure 4 (which will probably be figure 5) modifications are presented in red, but there is no detailed description of the modifications that are represented in figure 5, the authors should explain what the "c" and the circles highlighted in red represent as the proposed modifications. And such descriptions are expected to be close to the figure, or to refer directly to the figure in the text.
5) Authors should make a careful review of the equations presented, first equation 1 and equation 3 are the same, which implies that in equation 3 there is an error.
6) Equation 6 refers to the parameter number of the ASPP module, while equation 7 refers to the parameter number of the DASPP module, it would be advisable to use different notation, for example, in the subscript or superscript to denote the difference of both parameters in the equations.
7) Tables should not be separated, e.g. table 5 should be on one sheet.
8) In tables 5, 6 and 7 the speed parameter is presented in milliseconds per frame, although in many semantic segmentation works the speed parameter measured in frames per second is used, which allows the reader to evaluate if it can be used in real time, considering precisely the speed of some video sequences, so I suggest the authors to add this parameter.
Round 2
Reviewer 1 Report
The authors have satisfactorily addressed my previous comments.